# Network assessment and modeling the management of an epidemic on a college campus with testing, contact tracing, and masking

Gregg Hartvigsen [ID]*

Biology Department, SUNY Geneseo, Geneseo, NY, United States of America

* hartvig@geneseo.edu

**Data Availability Statement:** All code and data are available at https://github.com/GreggHartvigsen/Network-epidemic-model-college-campus.

**Funding:** The author received no specific funding for this work.

## Abstract

There remains a great challenge to minimize the spread of epidemics, especially in high-density communities such as colleges and universities. This is particularly true on densely populated, residential college campuses. To construct class and residential networks data from a four-year, residential liberal arts college with 5539 students were obtained from SUNY College at Geneseo, a rural, residential, undergraduate institution in western NY, USA. Equal-sized random networks also were created for each day. Different levels of compliance with mask use (none to 100%), mask efficacy (50% to 100%), and testing frequency (daily, or every 2, 3, 7, 14, 28, or 105 days) were assessed. Tests were assumed to be only 90% accurate and positive results were used to isolate individuals. The effectiveness of contact tracing, and the effect of quarantining neighbors of infectious individuals, was tested. The structure of the college course enrollment and residence networks greatly influenced the dynamics of the epidemics, as compared to the random networks. In particular, average path lengths were longer in the college networks compared to random networks. Students in larger majors generally had shorter average path lengths than students in smaller majors. Average transitivity (clustering) was lower on days when students most frequently were in class (MWF). Degree distributions were generally large and right skewed, ranging from 0 to 719. Simulations began by inoculating twenty students (10 exposed and 10 infectious) with SARS-CoV-2 on the first day of the fall semester and ended once the disease was cleared. Transmission probability was calculated based on an $R_0$ = 2.4. Without interventions epidemics resulted in most students becoming infected and lasted into the second semester. On average students in the college networks experienced fewer infections, shorter duration, and lower epidemic peaks when compared to the dynamics on equal-sized random networks. The most important factors in reducing case numbers were the proportion masking and the frequency of testing, followed by contact tracing and mask efficacy. The paper discusses further high-order interactions and other implications of non-pharmaceutical interventions for disease transmission on a residential college campus.

**Competing interests:** The author has declared that no competing interests exist.

# Introduction

There remains a great deal of interest in understanding and predicting the dynamics of the spread of the SARS-CoV-2 virus and similar infectious agents through populations. Analytical models are useful for estimating spread rates and extent of epidemics but lack the realistic structure of how people actually encounter each other. Network-based models, on the other hand, allow for discrete modeling of epidemics through more realistically-structured populations [1–3]. These models, however, usually use standard network structures to model viral spreading through populations (e.g., [3, 4]). The current work overcomes this by using actual enrollment data for a residential, liberal arts college with 5539 students.

A variety of models have been used to investigate potential spread and containment using different non-pharmaceutical interventions [5]. The results suggest that government-mandated lock downs, for instance, are essential to work toward reducing COVID-19's spread (achieving an $R_0 < 1.0$). However, the latter has been criticized for not incorporating the benefits from practices such as contact tracing [6, 7].

There are many studies that have demonstrated the effectiveness of different non-pharmaceutical interventions for the containment of SARS-CoV-2 among people. Masks, for instance greatly reduce the emissions of aerosolized droplets that are the leading cause of transmission [8, 9]. Additionally, testing and subsequent quarantining has been shown to be effective in reducing transmission rates [10, 11] and are having effects on other directly transmitted diseases [12]. In this paper we explore the interactive effects of COVID-19 testing, isolation, quarantining, and different proportions of people using masks that differ in efficacy within a real college network. The model relies on actual enrollment data in classes from a college with more than 5500 students.

Much remains unknown about the effectiveness of these interventions, such as masking [13]. In particular, there are differences between different types of masks, ranging from the common bandanna (neck gator) to N95 respirators [9, 14]. Because of this it is important to examine how masks with different efficacies might influence the spread of COVID-19 through a population. In addition, there are differences in the extent to which people use masks and wear them appropriately. In one study 86.1% of adults ranging in age from 18-29 chose to wear masks [15]. Despite this encouraging use of non-pharmaceutical interventions the pandemic has not been contained.

The US Centers for Disease Control and Prevention (CDC) has provided guidelines for institutions of higher education for safe operations [16]. Included in these recommendations are a range of practices from lowest risk to highest risk. The work here addresses this range by assessing what happens across the spec trum of safety, from the highest risk with in-person classes without non-pharmaceutical interventions to the lowest risk with no in-class meetings (simulated here through the use of masks that are 100% effective and used by everyone).

It is the hope of this work that we can better understand and predict the dynamics of infectious diseases and the effect on control measures in residential college communities.

# Methods

Anonymized college enrollment and residence data for 5539 students from a two-semester academic year (2019-2020) were acquired from the SUNY Geneseo Office of Institutional Research, with permission from the SUNY Geneseo Institutional Research Board [17]. This particular college, in addition to providing data to the author, is appropriate for this analysis because, being a relatively rural campus in western New York, USA, it is attended primarily by traditionally-aged students (18-21) who reside on or near campus. The campus reports a 20:1 student-faculty ratio. Additionally, it's size (5539 students) places it in the 87[th] percentile of all

US colleges and universities [18]). Because students are represented only by ID numbers and their majors there is no risk to their privacy. Additionally, the residence networks consist only of vertices and edges for those living together, without any other identifying information. Students living alone have no connections in the residential networks but do make connections across classroom networks. The daily class enrollment networks are made up of vertices and edges that connect student ID numbers when in the same class. These data were provided by the administration and include students that began the semesters. It is assumed no students leave school in the middle of either semester. Daily network sizes ranged from 3108 to 4919 vertices that were connected with between 109,000 to 305,000 edges. Multiple edges were permitted between vertices (e.g., students could be in two classes together and live with each other). No faculty or instructors were included in the networks. In addition, equal-sized random networks were created for each day for both semesters.

The degree distributions, average path lengths, and average clustering coefficients (transitivity) for the college and random networks for each day of the week for both fall and spring semesters were calculated.

An $SEIRI_{sol}Q$ network model (states include susceptible, exposed, infectious, recovered, isolated, and quarantined) was developed to simulate the spread of the SARS-CoV-2 virus through a population of undergraduate students (Fig 1). The network changed for each day of the week as students attended their various classes. On weekends students were assumed to only come into contact with their house mates. Fall and spring semesters were assumed to continue without interruption. At the beginning of the fall semester 20 students were assumed to begin classes infected with the virus (ten categorized as exposed and ten infectious). Students remained in the non-infectious exposed class for two days. After a 10 day infectious period ended individuals would enter a recovered state and could not be reinfected. The basic reproductive number ($R_0$) was set at 2.4 [19] which follows an earlier report which suggested the same rate [5] (see Table 1). These and additional parameters for the model are provided in Table 1. Simulations ended when the disease was cleared or there were no remaining susceptible neighbors of infectious individuals. The model assumes no individuals are able to become reinfected which has been found to be relatively rare [20]. Model, statistics, and network construction and analysis were completed using R [21] and the igraph package [22].

Simulations were run to compare spread under unmitigated conditions between the student and random networks (no masking or testing). Individuals were initially susceptible with 10 individuals randomly inoculated as exposed with an additional 10 individuals inoculated as infectious. Exposed individuals became infectious after two days and remained infectious for 10 days (see Table 1).

### Testing, contact tracing, and masking

All individuals were tested each semester but at different time periods. For each test cycle period students were randomly assigned days they would be tested. Time periods were 1, 2, 3,

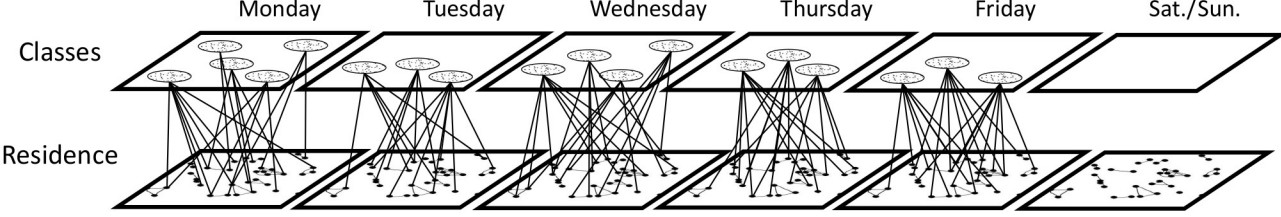

**Fig 1. Example of student networks over time.** Students (dots) in the each class (ovals in upper boxes) form complete networks. Students in residences form relatively sparse networks (below). Infected students carry their infections from day to day, even if not in a class on certain days.

**Table 1. Parameter settings for simulations.** These are the basic settings used in the simulations. This resulted in 6400 simulations.

| Parameter | Settings | Sources |
|---|---|---|
| $R_0$ | 2.4 | estimated |
| Number days exposed (latent) | 2 | [23] |
| Number days infectious | 10 | [24] |
| Network types | College, Random | |
| Test cycle (days) | 0, 1, 2, 3, 7, 14, 28, 105 | |
| Test results delay | 1 day | |
| Test accuracy | 90% | |
| Contact tracing | yes/no | |
| Number days in isolation/quarantine | 14 | CDC rec.** |
| Proportion population masked | 0, 0.5, 0.75, 0.861*, 1 | [25] |
| Mask efficacy | 0.5, 0.75, 0.95, 1.0 | |
| Number initially exposed/infectious | 10/10 | |
| Number of replicates | 10 | |

Table notes:

* for 18-29 year olds [25].

** At the time of writing CDC recommends a 10 day quarantine period.

7, 14, 28, or 105 days. A one means students were tested daily and 105 means students were tested once per semester. In addition, simulations were run without testing for infectious individuals (see Table 1). Students were not tested if they were currently awaiting test results or in either isolation, quarantine, or recovered. Test results were evaluated one day after testing with a 90% positive accuracy rate (false positives were not considered). If testing was being used then students that were infectious at the time of the test were isolated for two weeks. If contact tracing occurred then all susceptible neighbors were quarantined for two weeks. The model treated quarantine as complete with no contacts allowed. COVID-19 tests of students that were in the exposed state when tested were considered negative and were returned to the network and allowed to move to the infectious state. After 10 days in the infectious state students were moved into a recovered class and could neither receive nor share the virus with neighbors.

The transmission probability ($T_{d,s}$) was calculated for each day of the week ($d$) for each semester ($s$). This probability was used to determine the likelihood that an infectious individual would pass the virus to a susceptible neighbor on a given day. $T_{d,s}$ was determined using the following relationship:

$$T_{d,s} = 1 - \left(1 - \frac{R_0}{K_{d,s}}\right)^{(1/D_I)} \tag{1}$$

where $K_{d,s}$ is the median degree of the network on day $d$ of each semester ($s$) [4]. Individuals were assumed to be infectious for 10 days ($D_I = 10$, [24]). On weekends degrees were low leading to relatively high transmission probabilities, which effectively simulates close interactions among roommates. This relationship for $T_{d,s}$ results, on average, of infectious individuals infecting $R_0$ susceptible neighbors in a completely susceptible neighborhood. However, as the infection spreads the realized spread rate generally decreases as the number of susceptible neighbors of infectious individuals decreases.

## College Network

## Random Network

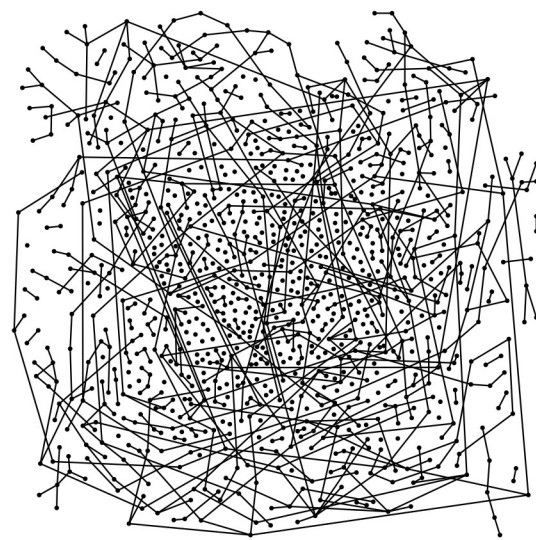

**Fig 2. College and random housing networks for the fall 2019.** The network on the left is a random sample of 1000 of the 4919 vertices and the 664 associated edges showing the structure of the housing network. The network on the right is a random network with the same number of vertices and edges. Discrete housing communities are apparent in the college network (left) and absent in the random graph (right). Vertex arrangements were completed using the Davidson-Harel layout algorithm.

## Results

We begin with a comparison evaluating the structural differences between the college and random networks. This includes the degree distributions and clustering coefficients as well as the average path lengths for all students and grouped by majors. These metrics play important roles for the overall dynamics of disease transmission. This is followed by a discussion of the results from simulating disease transmission through these networks.

### The college network structure

The college networks include course enrollments for Monday–Friday plus the housing data for all days of the week over both fall 2019 and spring 2020 semesters. Students were assumed to interact only with their housemates on weekends. These networks, were strikingly different from the random networks for each day of the week and between semesters (see Fig 2). These structural differences led to significant differences in the dynamics of disease spread between the college and random networks (discussed below). The housing network includes 4930 students (89%) with the remaining 609 students not reporting their off-campus residences. A majority of students can be seen living in pairs (dyads in Fig 2). Undoubtedly, weekend and evening gatherings could contributed substantially to epidemic spread.

**Degree.** The number of connections (degree) for individuals in the college networks ranged from 0 to 719 and varied from day to day and were right skewed (Fig 3), with most individuals having a total degree less than 100 each day. A small number of students had a zero degree on a day in which they had no classes and happened to not have their housing location reported. On the weekend ("SS" in Fig 3) we can see that a few individuals had degrees greater than 10. The most frequent degree was just one, forming dyads (note that in Fig 3 a one was added to all degrees).

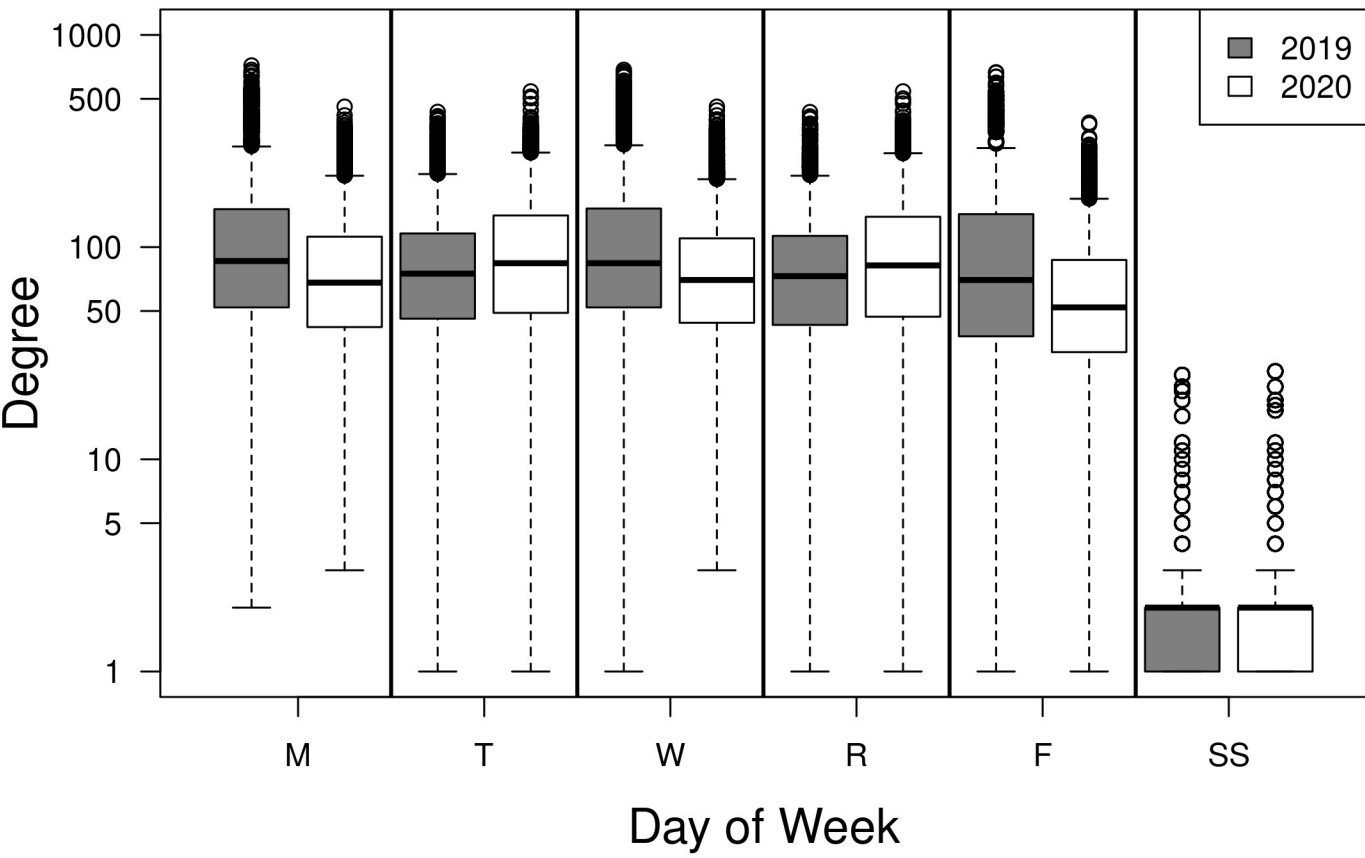

**Fig 3. Degree distributions for college networks by day for fall 2019 and spring 2020 semesters.** Weekday networks include both enrollment and housing connections for both the fall 2019 and spring 2020 semesters. The weekend networks ("SS") include only the housing connections. Degree values all were increased by one and shown on a semi-log plot.

**Average path length.** The average path length (APL) is a metric that summarizes the average number of steps from each student to all other students through both the enrollment and housing networks. APL varied by day of week (Fig 4). Most notably, students are very highly connected with fewer than three steps separating students, on average. Some differences are apparent between semesters, particularly between MWF and TR classes. Average path lengths are longer in the college network compared to the random network due to clustering that takes place within majors. Additionally, we can see that students in different majors had variable average path lengths with no clear pattern related to size of major (Fig 5).

**Clustering coefficient.** The clustering coefficient (CC), or transitivity, for a vertex in this analysis is the average number of triangles formed by neighbors divided by the total possible number of triangles. High average values for this coefficient suggest students who are clustering together in groups. In the college networks clearly students are gathering in classes which form complete subgraphs. This measure can play an important role in the spread of a disease within a group but also can function to isolate different groups from each other. The college networks are clearly highly clustered compared to the random networks and serves as an important metric differentiating the college from random networks (Fig 6).

### Epidemic dynamics

Simulations were begun on the first day of classes in the fall 2019 semester. For each replicate simulation on both the college and random networks 20 randomly selected students were

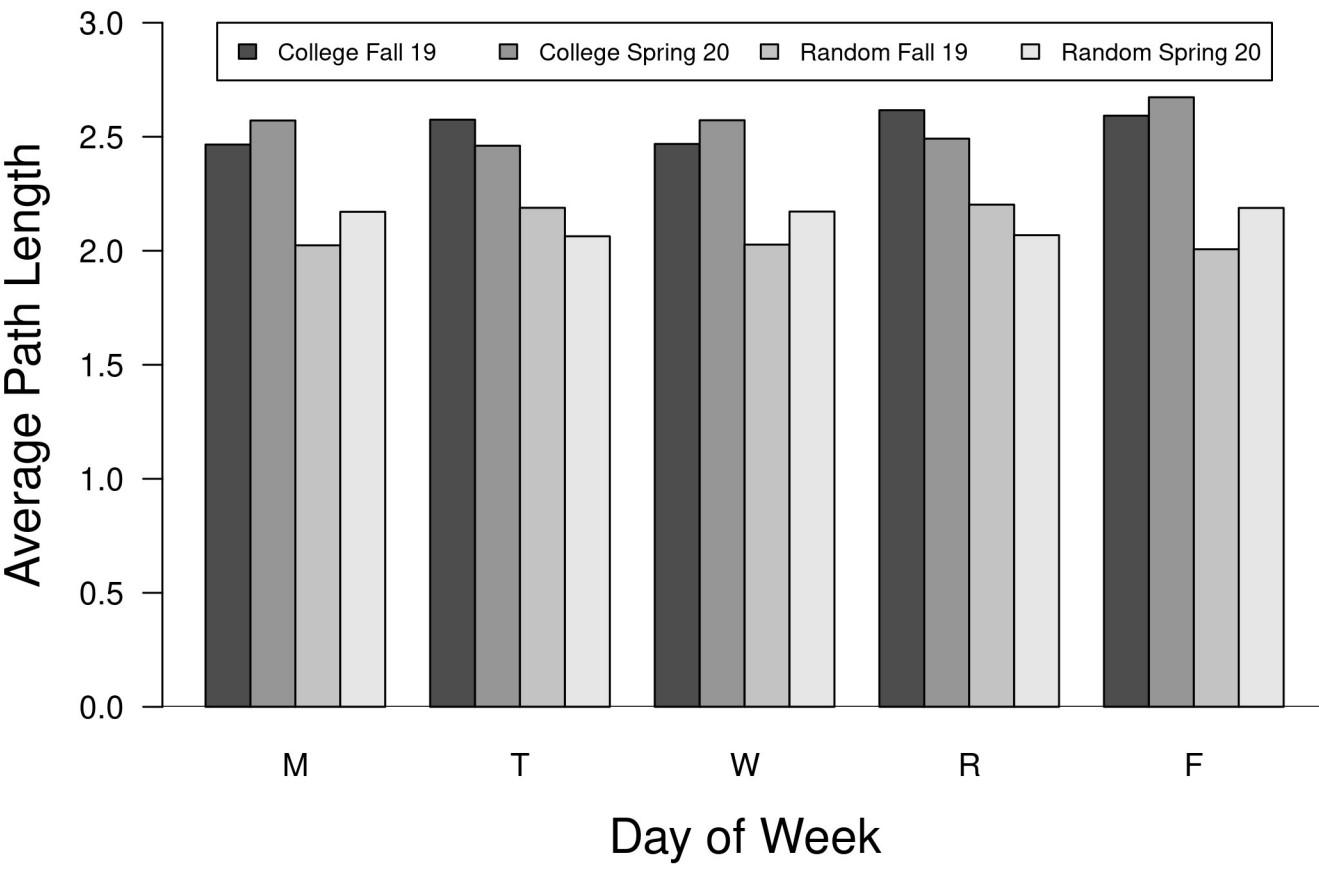

**Fig 4. Average path length for college and random networks.** The average path length (APL) for weekday networks from the fall 2019 semester are shown. Additionally, APL values for random networks also are shown, which are consistently lower.

inoculated with SARS-CoV-2 virus. Half were placed in the exposed class and half in the infectious class. The disease always spread on both the college and random networks (Fig 7). The dynamics exhibited periodicity on a weekly schedule due to restricted spread on weekends when individuals in the model mix only with those with which they reside. As a result, relatively few individuals become exposed over weekends.

All five factors (see Table 2) significantly influenced the number of students becoming infected individuals in the network populations. Most important of these are the frequency of testing (test cycle) and the proportion of students using masks (Fig 8). These two factors explained a combined 45 percent of the total variance in the model and, as testing and masking levels increased the number of cases fell (Table 2). Additionally, all higher-order interactions (three-, four-, and five-way interactions among the factors) were statistically significant (not shown in Table 2). Testing, and subsequent isolation and quarantining of contacts, significantly reduced the total number of infections, accounting for a combined 27% of the overall variance. The effect is quite large for even small levels of testing and subsequent isolation of individuals who test positive, followed by quarantining of neighbors. There was a significant reduction in the numbers of individual infected with as little as 1% of the students tested daily (students testing only once per the 105 day semester, Fig 8).

In the absence of interventions (testing and masking) disease prevalence reached its highest levels seen in all simulations. Overall, the college, with its higher APLs and clustering

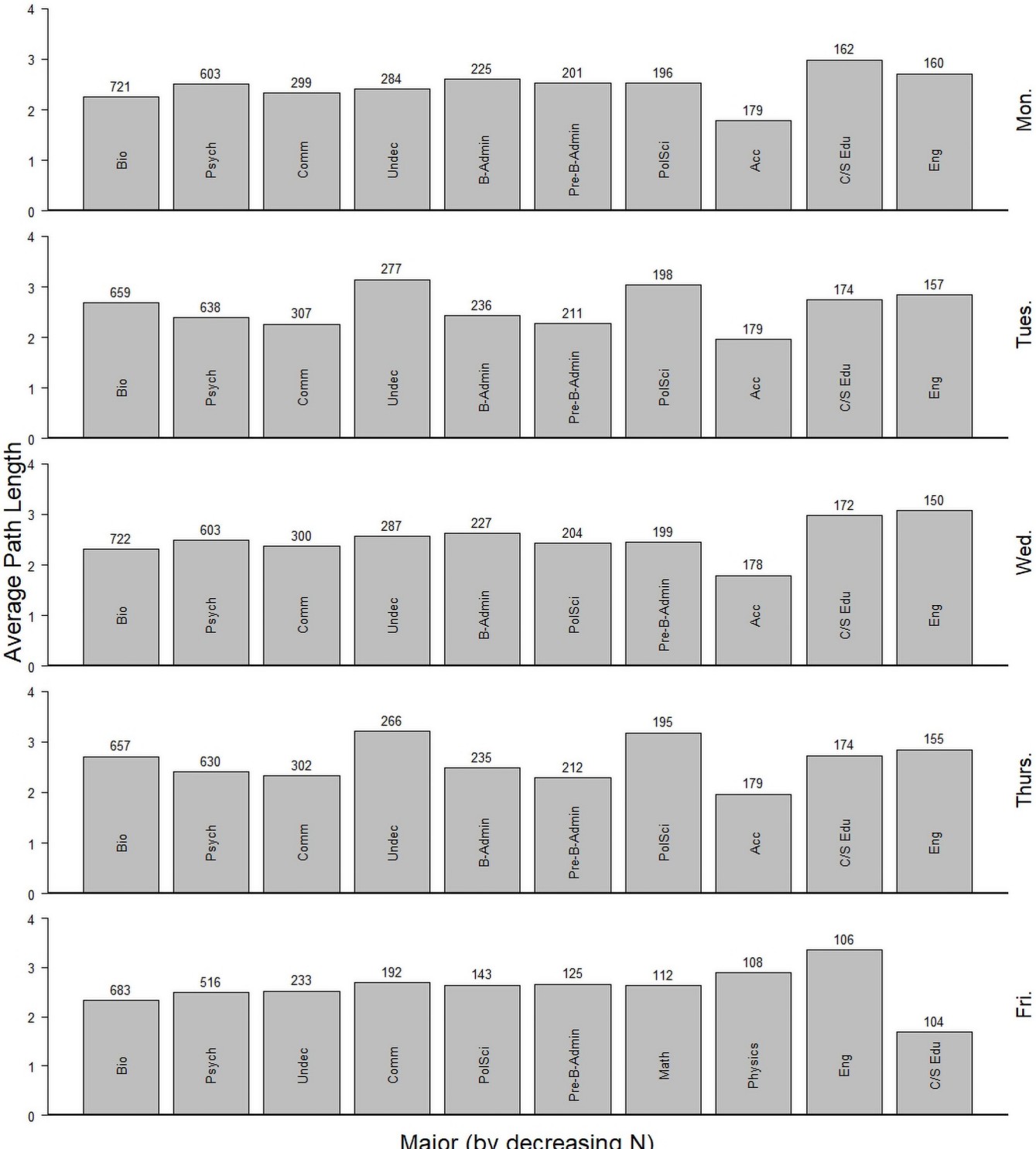

**Fig 5. Average path length for students separated by major and weekday.** The average path length (APL) for students in the 10 most populous majors for each weekday during the fall 2019 semester are shown. Numbers above bars represent the number of students registered in a class on that day by major. Abbreviations are: Biology (Bio), Psychology (Psych), Communications (Comm), Undecided (Undec), Political Science (PolSci), Business Administration (B-Admin), Pre-Business Administration (Pre-B-Admin), Accounting (Acc), Childhood/Special Education (C/S Edu), English (Eng), and Mathematics (Math).

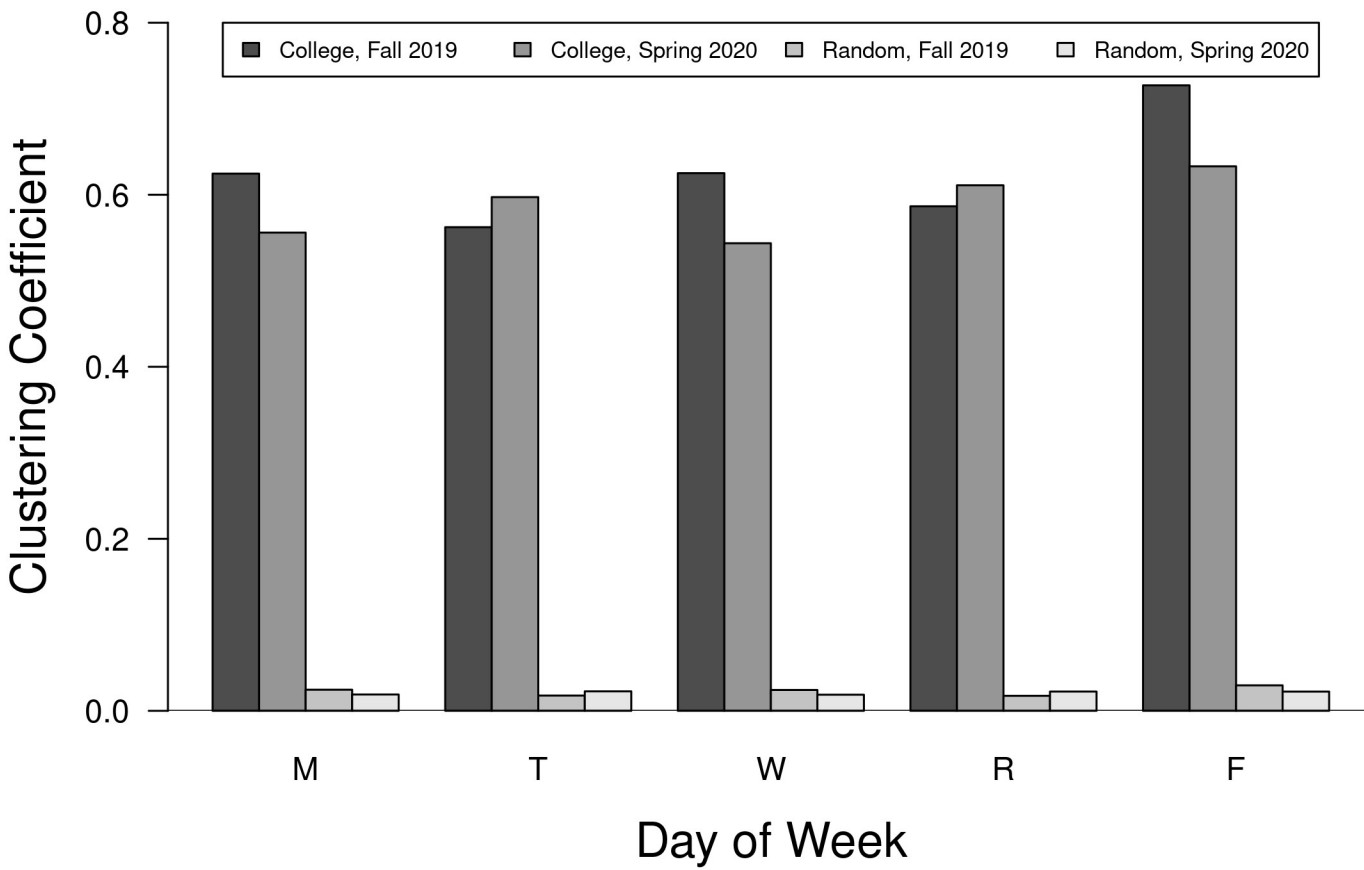

**Fig 6. Clustering coefficients for weekday college and random networks.** The global clustering coefficients (transitivity) for the college and one rendition of the paired random networks based on the same number of vertices and edges.

coefficients, exhibited fewer infections than seen in the random networks (Table 3). Additionally, the epidemics on the college networks reached their peaks and ended later than those on random networks and resulting in fewer infected individuals. An additional challenge for college health providers is the result that the epidemic peak on this campus occurred after just one month and involved over 25% of the student population.

## Testing and contact tracing

Increasing testing frequency with contact tracing significantly reduced the number of infections. Previous work suggesting that testing every other day minimized the number of infections [26]. Results here concur (Fig 8) although, numerically, fewer individuals were infected by testing daily. There was no significant difference between the number of infected individuals for testing rates of every day to just once every two weeks. However, from an implementation standpoint the number of tests needing to be administered in the college network population was highest when tests were administered every other day (Fig 9). This is due to the high reproductive rate of SARS-CoV-2 ($R_0 = 2.4$) that allow the virus to spread more than when testing occurred daily. The fewest tests were required when testing was conducted daily and when done just once per student per semester, although the latter was the least effective method for controlling COVID-19 (Fig 8).

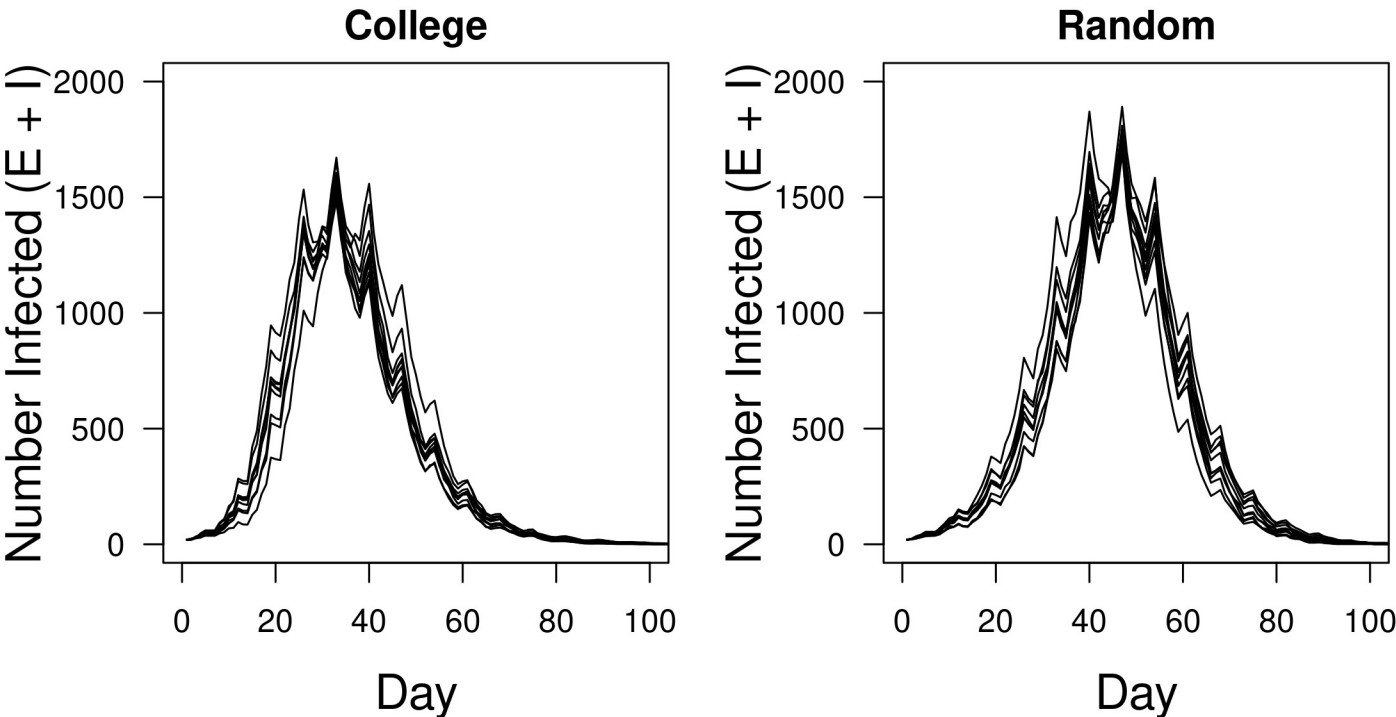

**Fig 7. Time series dynamics of the simulation.** The number of exposed (E) and infectious (I) individuals per day for ten replicate simulations on the college (left) and random (right) networks. Fewer individuals become infected on the college network than the random network because of the community structures (longer APLs and higher clustering coefficients) within majors. The weekly cycles are caused by the increased but sparser connectivity on weekends.

**Table 2. ANOVA table for number of infections.** The main effects and two-way interactions are shown as percentages of the sums of squares from the ANOVA. All factors and higher-order interaction terms were highly statistically significant (P < 0.001). The main and two-way interactions (shown) accounted for 88% of the variance (overall adjusted $R^2$ = 0.995).

| Factor | df | Percent SS | p |
|---|---|---|---|
| Test Cycle | 7 | 22.24 | < 0.001 |
| Contact Tracing | 1 | 5.00 | < 0.001 |
| Proportion Masked | 4 | 23.33 | < 0.001 |
| Mask Efficacy | 3 | 1.97 | < 0.001 |
| Network Type | 1 | 0.36 | < 0.001 |
| Test Cycle x Contact Tracing | 7 | 3.75 | < 0.001 |
| Test Cycle x Proportion Masked | 28 | 19.29 | < 0.001 |
| Contact Tracing x Proportion Masked | 4 | 7.94 | < 0.001 |
| Test Cycle x Mask Efficacy | 21 | 3.19 | < 0.001 |
| Contact Tracing x Mask Efficacy | 3 | 0.41 | < 0.001 |
| Proportion Masked x Mask Efficacy | 12 | 0.77 | < 0.001 |
| Test Cycle x Network Type | 7 | 0.06 | < 0.001 |
| Contact Tracing x Network Type | 1 | 0.26 | < 0.001 |
| Proportion Masked x Network Type | 4 | 0.17 | < 0.001 |
| Mask Efficacy x Network Type | 3 | 0.09 | < 0.001 |
| Residuals | 5760 | 0.95 | < 0.001 |

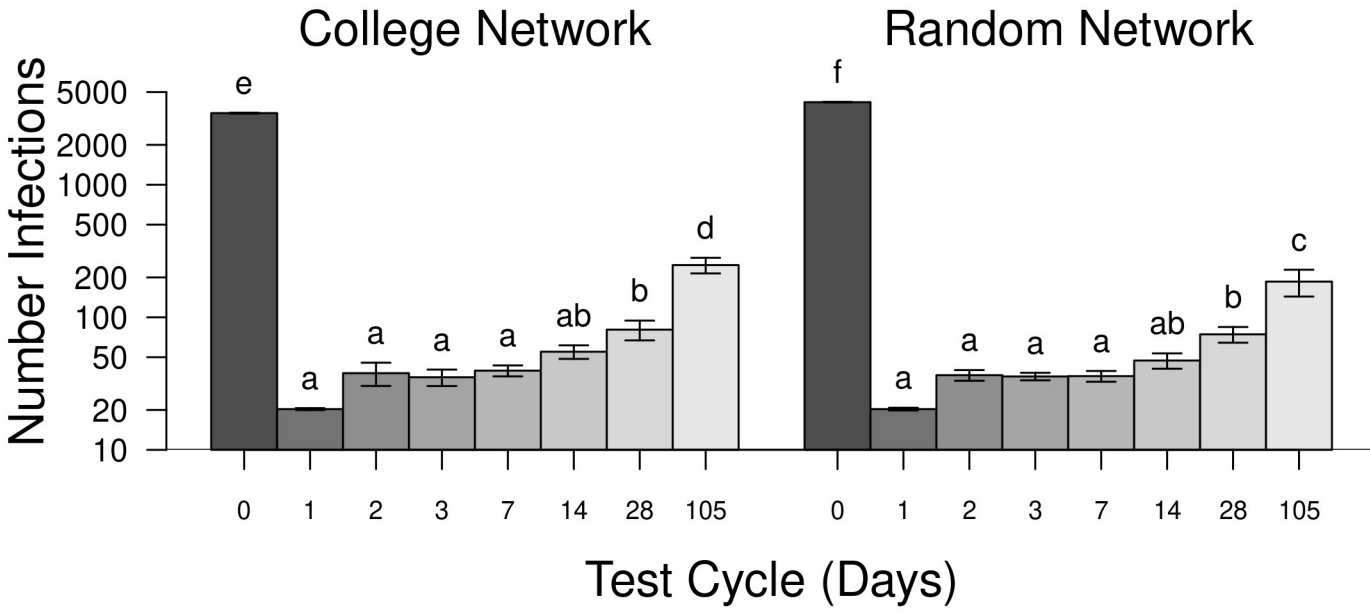

**Fig 8. The number of infections for different testing frequencies for both networks.** The average number of infections for simulations with testing and contact tracing at different intervals. Note that test cycles are the days between testing for individuals. The largest number of infections occurred when no testing was done (0 days), followed by testing students only once per semester (105 days). Error bars are ± 95% confidence intervals. Samples sharing letters are not statistically different.

An additional 19% is explained by the interactive effects of the number of people tested and the proportion of individuals that mask, regardless of the efficacy of masks, which ranged from 50% to 100% effective.

## The effects of masking

Masking significantly reduced the number of infected individuals in these populations. Importantly, There was a significant interaction between the proportion of people masking and the efficacy of the masks (Fig 10, Table 2). As can be seen in this figure, changing from no masking to even using masks that are 50% effective at blocking the transmission greatly reduced the number of infections.

**Table 3. Comparison of unmitigated spread on the college vs random networks.** The four main response variables are presented as means (± 95% CI). No masking or testing was done. All eight samples were normally distributed. All responses were significantly different between the college and random networks using a t-test (df = 18). Note that the outbreaks were completed after approximately one semester (105 days) and infected an average of 63% and 76% of the college and random network individuals, respectively.

| Metric | College | Random |
|---|---|---|
| Total number of infections* | 3467.6 (36.5) | 4197 (32.4) |
| Duration of epidemic* | 100.5 (3.9) | 107.9 (5.5) |
| Peak number of infections* | 1656.7 (50.7) | 1799.0 (38.5) |
| Day of peak infections** | 33.0 (0) | 46.3 (2.8) |

Differences are statistically significant (* $p < 0.001$, ** $p = 0.024$).

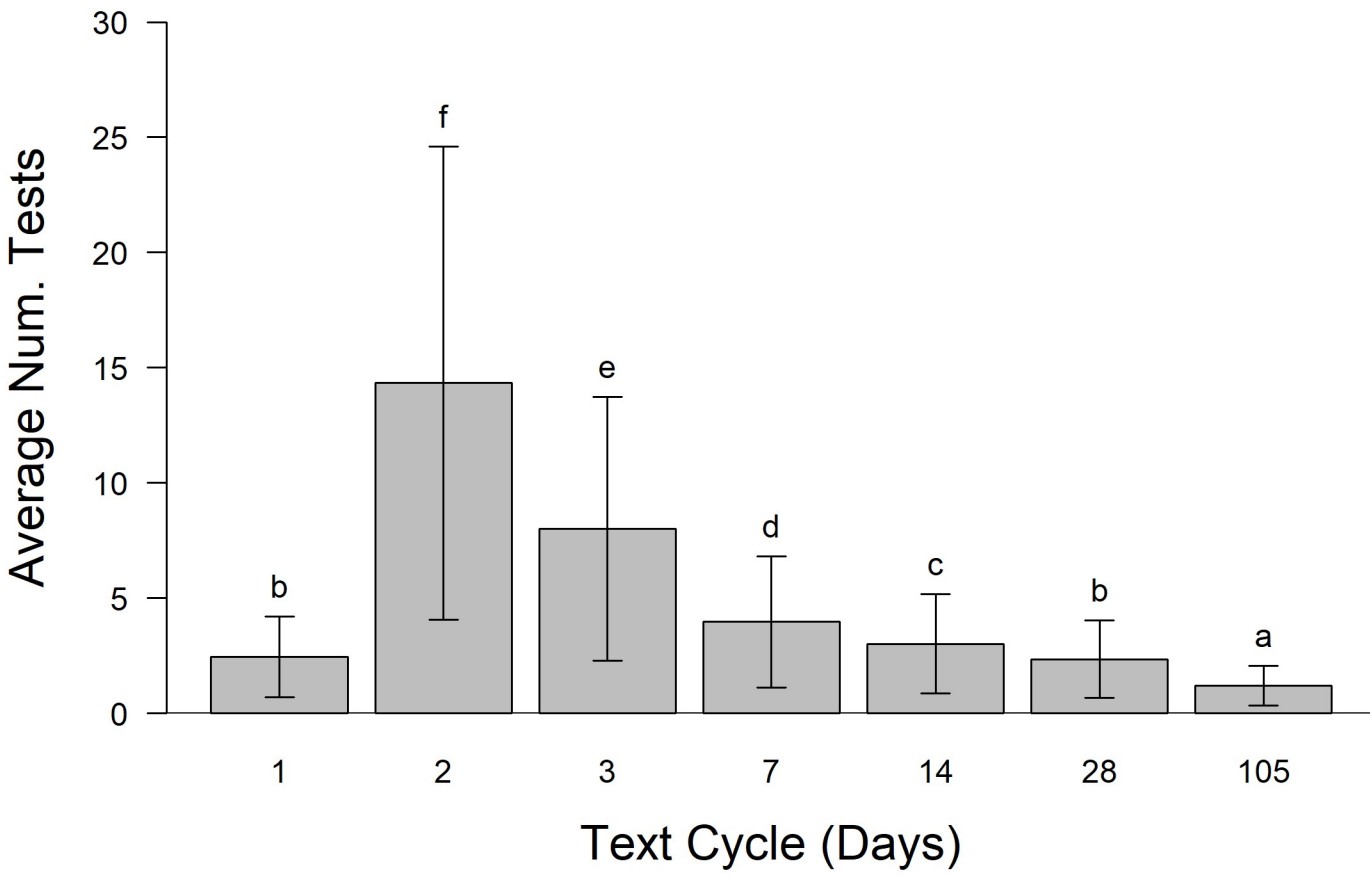

**Fig 9. The average number of tests conducted per person versus testing cycle.** These simulations include only tests with contact tracing and no masking using the college networks. The average number of tests per person was greatest when individuals were tested every other day. This was far greater than when individuals were tested daily because individuals that tested positive were isolated, not tested during this period and most effectively curtailed the outbreak. Error bars are ± 95% confidence intervals. Samples sharing letters are not statistically different.

## Which students contract COVID-19?

Unfortunately, essentially all the students are vulnerable and likely to contract COVID-19 based on how students are connected through coursework and their residences. Having the majors of students allows us to determine whether students of different majors are more or less likely to contract COVID-19. In Fig 11 we can see, after creating induced subgraphs by major, there is no relationship between the mean proportion of students contracting the disease agent and average path length (A), mean degree (B), or clustering coefficient (C, Fig 11). However, there was a weak, positive relationship between proportion of students infected and the size of the major (panel D, $F = 4.56$; $df = 1, 38$; $p = 0.039$; $y = 0.079x + 0.457$; $R^2 = 0.084$). Additionally, there was a positive, non-linear relationship between the sizes of individual classes and the proportion of students that got infected, although this relationship is poorly modeled with any simple asymptotic function. Also, the relationship appears important only as class size exceeds about 50 students (Fig 12).

## Discussion

To predict the dynamics of a disease like COVID-19 spreading through a college campus, or any population, it is helpful to include the actual structure of the population. This work relies

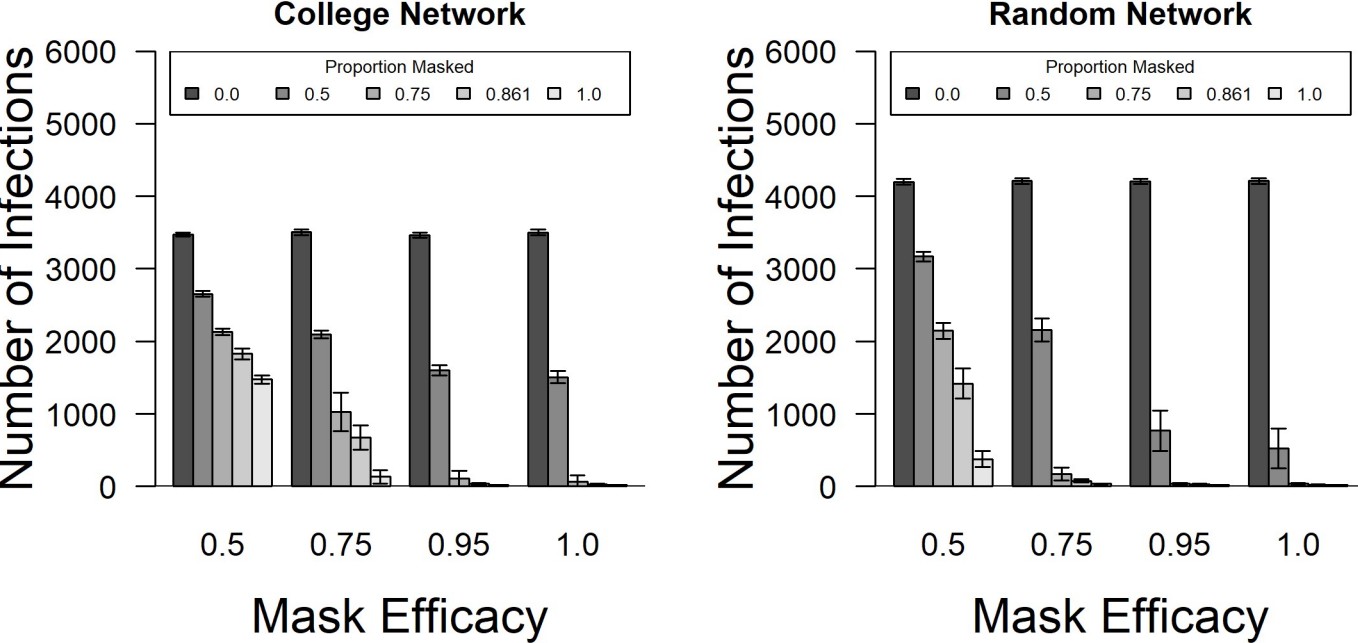

**Fig 10. Comparison of the number of infected individuals in response to masking levels and efficacies between networks.** The number of individuals that became infected under different levels of masking and mask efficacy for both the college and random networks. This three-way interaction was statistically significant (F = 74.14; df = 12, 5760; p < 0.001) along with the two-way interactions within network types. Without masking more students contracted the disease on the random networks but masking proved less effective, comparatively, on the college network due to the high clustering of students. Error bars are ± 95% confidence intervals.

on class enrollment and housing data from a college with 5539 students. The daily course enrollment and residential network structures were analyzed and used as frameworks on which to simulate the spread of the disease through the population. These networks and model dynamics also are compared against those responses in a set of same size, randomly constructed networks.

When a disease, such as COVID-19, enters a college population there exists a variety of challenges to minimizing its spread. This work investigated the effects of various methods to minimize disease spread. The findings suggest that the risk of disease spread is reduced significantly by the actual structure of students who are non-randomly enrolled in classes, mainly with members of the same majors. This appears to be largely due to longer average path lengths and higher clustering coefficients found among students in the college networks compared to random networks. The longer a path is between students the less likely a transmissible disease agent will be successfully transmitted. Interestingly, there was a significant, although weak, relationship between the size of majors (number of students) and the number of infections. Additionally, there was clearly an increased risk of students contracting a disease like COVID-19 from larger classes.

Admittedly, entering into a non-voluntary 14-day quarantine period is disruptive to everyone, particularly college students. In this model, too, students were assumed to be completely compliant during the quarantine period. When implementing frequent testing of students in the model many students ended up in quarantine. This greatly disrupts learning environments. There is evidence that shorter quarantine periods may be effective [27]. Additionally, this model assumes a relatively rapid turn around on testing results (1 day). Interestingly, testing students every day resulted in very low numbers of tests administered because infectious individuals were rapidly identified and isolated, despite test accuracy being only 90% (see Fig 9).

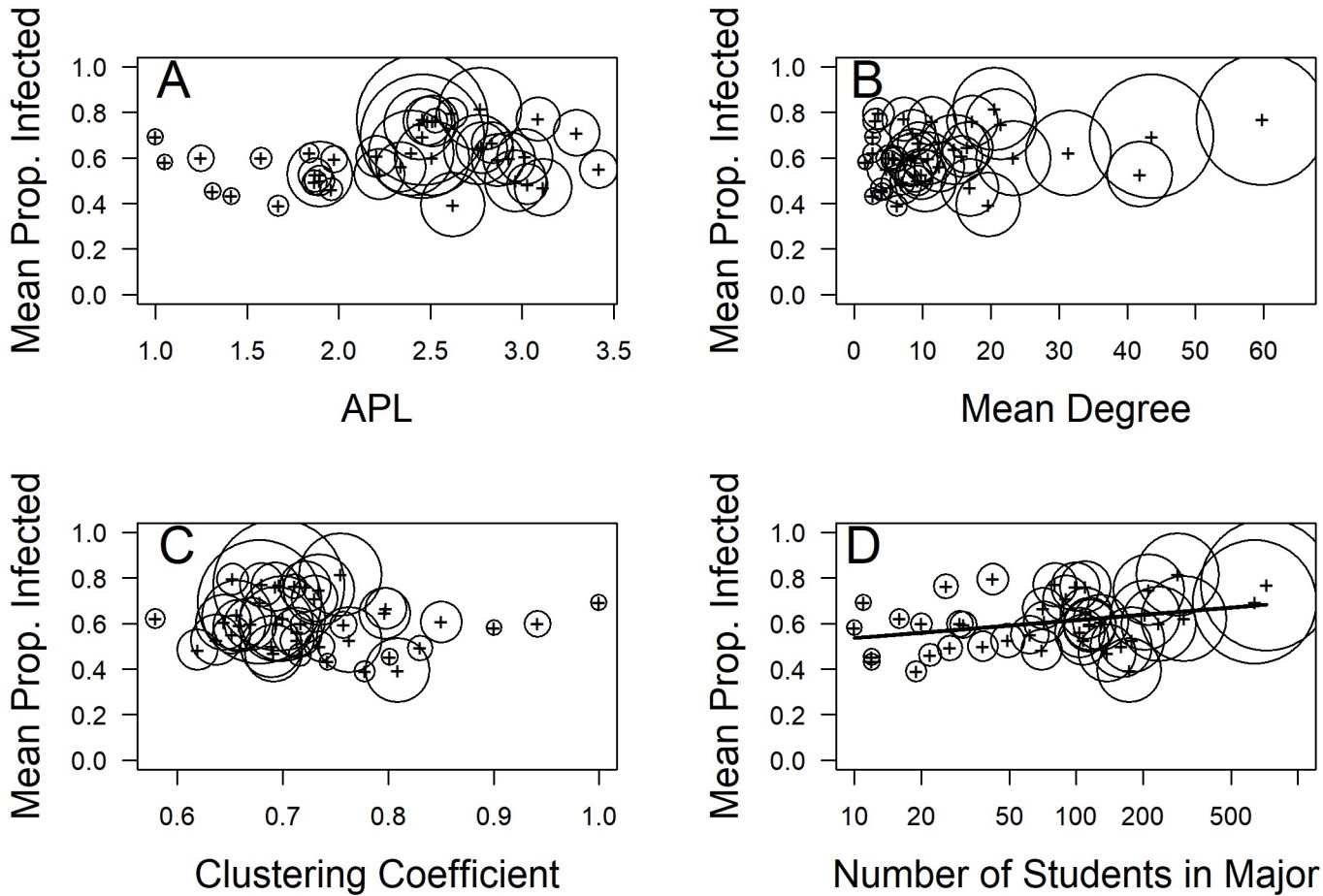

**Fig 11. Testing factors associated with the proportion of students that contracted COVID-19 by major.** Circles with dots in their centers represent different majors. Circle areas are proportional to the number of students in those majors. Simulations were replicated five times for the proportion infected. Network metrics include average path length (A), mean degree (B), average clustering coefficient (C), and the number of students in each major (D). The proportion of infected students is positively related only to the number of students in the major (F = 4.56; df = 1, 38; p = 0.039; y = 0.0789x + 0.457; $R^2$ = 0.084).

Daily testing also resulted in the fewest number of cases. However, the administration of daily tests for all students on a campus is likely challenging. Modeling by Paltiel et al. (2020) concluded that testing on a college campus would be optimal at, in decreasing order of effectiveness, 2-, 1-, and 7-day intervals [26]. Results here suggest that there would be an approximately five-fold increase in the number of tests required when implementing a two day testing period over testing daily.

Students in larger classes tended to have a higher risk of becoming infected. This leads to a recommendation that class size be kept as low as is feasible. However, with the use of even low efficacy masks the epidemic can be well contained.

The current work does not model the effects of social distancing as an effective non-pharmaceutical intervention. Clearly, the level of concern by students can vary greatly and this was accounted for through the testing of mask efficacy (from 50–100% effectiveness). This model calculates a transmission probability for all students each day. With this approach it is assumed that students all share the same likelihood of either infecting others or being infected by others. It would be interesting to know how important these assumptions are in affecting the outcomes reported here.

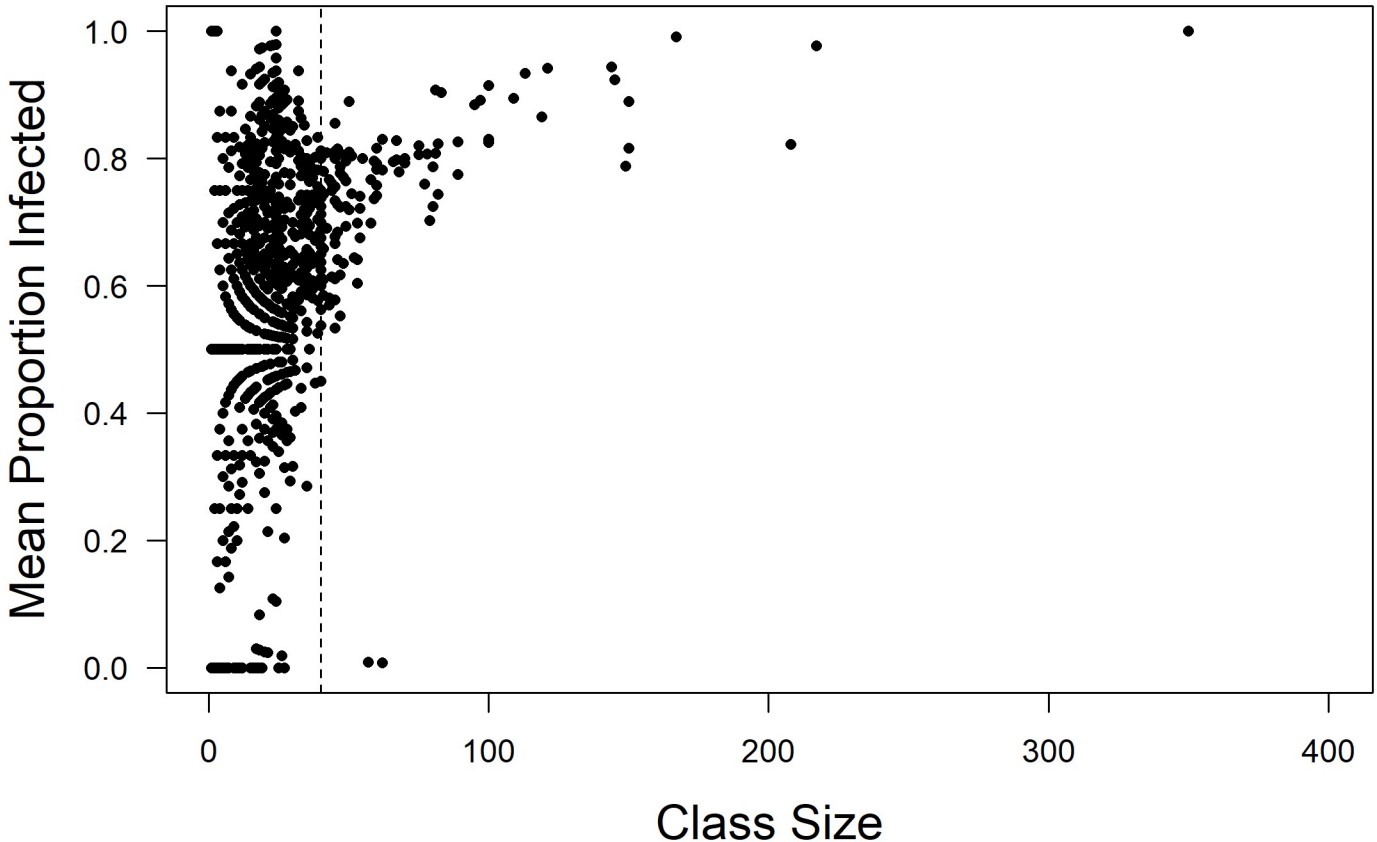

**Fig 12. Mean proportion of students infected versus class size.** The mean proportions of students infected are from all 1458 classes in the fall 2019 semester. Two simulations were run without interventions (e.g., masking or contact tracing) and the proportions infected were averaged. The results suggest that large class sizes impose increased risk of infection, particularly classes over 40 students (vertical dashed line). Although there are relatively few large classes these do represent a large number of students. Data are from the fall semester when most infections occurred.

Despite the complexity of the model with five factors investigated (see Table 2) interacting within a real college network with individual course schedules and residential housing, the model assumes individuals do not differ by race, sex, gender, socio-economic backgrounds, or age. Additionally, the model assume transmission of the virus occurs only in classes and residences. Although the number of tests was quantified the actual cost to a college may make testing difficult. The model also does not incorporate faculty or staff. It is possible that faculty would transfer a viral pathogen across classes or meet students during office hours, for instance. The model also does not address the challenges of variability in either morbidity or mortality, although it does assess conditions under which these are minimized. More recently, too, and not considered in this model, is the advances of vaccines that can protect students from contracting the disease, although these function similarly to effective masking by reducing the rate of infection. Nonetheless, the framework is novel in accounting for known interactions among students and quantifies the enormous value provided through the use of non-pharmaceutical interventions, including wearing masks, testing for infection, and subsequent follow ups with isolation of infectious individuals and quarantining of immediate contacts.

## Conclusion

Students on a college campus generally reside in very well connected networks by attending classes and living in residence halls. With a high reproductive rate for a directly transmitted

disease agent, such as SARS-CoV-2 students attending a residential college are at a high risk of spreading and contracting such a disease. Interestingly, the structure of the college network itself was sufficient to reduce the spread of the disease agent. This was due to students in majors taking classes together, leading to increased average path lengths and higher clustering coefficients compared to randomized networks. Additionally, wearing masks and utilizing frequent testing and contact tracing that leads to isolation and quarantine can greatly reduce spread in such a community. Even the use of poorly functioning masks alone greatly reduced transmission. Finally, class sizes greater than 40 students resulted in high proportions of those students contracting the disease. Therefore, working to manage the size of classes would likely reduce the incidence of transmissible diseases on college campuses.

## Author Contributions

**Conceptualization:** Gregg Hartvigsen.

**Data curation:** Gregg Hartvigsen.

**Formal analysis:** Gregg Hartvigsen.

**Investigation:** Gregg Hartvigsen.

**Methodology:** Gregg Hartvigsen.

**Project administration:** Gregg Hartvigsen.

**Resources:** Gregg Hartvigsen.

**Software:** Gregg Hartvigsen.

**Supervision:** Gregg Hartvigsen.

**Validation:** Gregg Hartvigsen.

**Visualization:** Gregg Hartvigsen.

**Writing – original draft:** Gregg Hartvigsen.

**Writing – review & editing:** Gregg Hartvigsen.

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
