## [Decision Letter · Decision Letter 0]

21 Jul 2021

PONE-D-21-17066

Network assessment and modeling the management of an epidemic on a college campus with testing, contact tracing, and masking

PLOS ONE

Dear Dr. Hartvigsen,

Thank you for submitting your manuscript to PLOS ONE. After careful consideration, we feel that it has merit but does not fully meet PLOS ONE’s publication criteria as it currently stands. Therefore, we invite you to submit a revised version of the manuscript that addresses the points raised during the review process.

Specifically, both reviewers ask for more details, like the motivation for studying COVID-19 transmission in a college setting, why only undergraduates are included in the study and not graduate students and faculty members, layouts of the dormitories and classrooms, and why the focus on mask wearing and testing, as opposed to hand washing or social distancing. Given the narrow focus of the study on the college setting, both reviewers also asked for the inclusion of a limitations section. Please address these comments, as well as other comments by the reviewers.

We look forward to receiving your revised manuscript.

Kind regards,

Siew Ann Cheong, Ph.D.

Academic Editor

PLOS ONE

Journal Requirements:

Reviewers' comments:

Reviewer's Responses to Questions

**Comments to the Author**

1. Is the manuscript technically sound, and do the data support the conclusions?

Reviewer #1: Partly

Reviewer #2: Yes

2. Has the statistical analysis been performed appropriately and rigorously? 

Reviewer #1: Yes

Reviewer #2: Yes

3. Have the authors made all data underlying the findings in their manuscript fully available?

Reviewer #1: Yes

Reviewer #2: Yes

4. Is the manuscript presented in an intelligible fashion and written in standard English?

Reviewer #1: No

Reviewer #2: Yes

5. Review Comments to the Author

Reviewer #1: Overall the manuscript provides important information about the importance of mask efficacy and testing, however, now that the vaccine has become readily available and the mask mandate for individuals who are vaccinated is no longer applicable this paper seems a little untimely based on the arguments presented by the author. Mask efficacy and testing can still be an effective non-pharmaceutical intervention for large gathering areas that are sustained for long periods, such as a college campus, where vaccination rates may vary and transmission is much easier. However, the paper would benefit from incorporating a more time sensitive analysis that includes some discussion of pharmaceutical intervention (i.e., vaccination) and why that was not included in the work shown here. Additionally, I would encourage the author to make major grammatical edits and use a third-person narrative instead of a first-person one.

Introduction

I think overall this just needs to be re-vamped more from a grammatical standpoint then content; overall the points feel slightly behind the times of where we are at with COVID-19 interventions, additionally more emphasis could be put on why you are looking at a college campus and the significance. I’m unsure why it needs to be a college campus compared to another high traffic area, or why you are focusing on this group as well.

Line 7-8: Explaining this famous network is an Influenza model, it’s easy to point the reader to the citation however it becomes a more accessible paper to know that this model is based on a virus transmission.

Line 10: Is 5539 people a medium sized college? Seems more worth while to just state the number instead of this abstract idea of “medium”

Line 11-15: was this work previously done before the era of contact tracing being utilized as a major public health intervention? I understand the background presented here but it’s worded in a way that is confusing.

Methods

Can the author explain the make up of the students and possibly the rationale as to why only undergraduate students where looked at and do not include graduate degrees; a rational as to why graduate students and faculty were not included is needed, as these are people that come into contact with undergrad students and could be links in transmission chains.

Additionally, what is the layout of the residential university housing?

How many people are on a dorm floor, are there shared facilities (shower, laundry, dinning etc.), is it one building for all the students? These probably don’t impact the numbers but is important when folks are generalizing results to similar colleges across the US.

Why did you only focus on masking and testing? Why not also look at hand washing or social distancing as a non-dharma intervention?

Results

Line 119: What is the n for the 89%? Please add that number.

Line 150: What are the five factors again? Would remind the reader

Line 163: Please define “higher-order interactions” for the reader

Section, Which students contract COVID-19: This section points out that there is little relationship about the characteristics described in this paper with transmission across students, this demonstrates a major limitation of the work where there is no discussion about the variations across race/gender, if someone is working on or off campus or at all, apart of greek life and other sociodemographic markers as well. I understand the convenience of pulling the data but there are some serious generalization issues that are not addressed in the discussion section.

Discussion

As noted above this paper needs a limitation section; I would recommend discussing the inability to analyze occupation/socio-economic status, cost and resources need to do every day or every other day testing, inclusion of graduate students, professors or other on campus staff they’d encounter, and not including vaccinated folks in addition to the briefly mentioned limitations.

I am confused by lines 240-242, as a reader I am under the impression that all students are being tested every day regardless of symptoms but this line states that there were low numbers of test administered every day, so are they not really tested everyday? The statement is contrasting to the methods and my interpretation of how testing interventions work in your model.

Reviewer #2: An interesting article based on the real world school networks of students in the context of SARS-CoV2.Its provides insights on the contact patterns in an education setting in US and provides some take away points based on simulations. However there are few concerns which need to be addresses.

1) The author uses subjective reference (I found, I analysed etc) a lot through the text. I guess this would not be fit for this particular journals style.

2) Provide some background details ion the school setting based on which the network data was constructed. Why it was selected, the location, type of students , etc. Data quality obtained from there etc. How missing information were addressed. ethical compliance etc.

3) Provide details/references for parameters which were used for epidemiological transition (R0, Mask protection etc). Are they based on real work epidemiological studies or assumptions?

4) There is no reference towards the assumed frequency of students interaction or distance between them in the class room. These considerations into the parameters could be influencing. This is important considering the transmission differences based on interaction intensity and assumed distance between which in the context of SAR-CoV2

5) the authors says that larger path length reduces transmission and larger classes might result in more transmission. need more discussion and references on this. could this be discussed in detail based on other publications (Per Block, Nature Human Behaviour 4, 588-596 2020).

6) The authors could provide a more clearer network diagram. Rather than providing the whole network, a clique or component with larger and smaller path length could be shown. the present ones dont would hardly make sense for the reader

7) provide the practical implications of this more detailly and also the limitations of the method

6. PLOS authors have the option to publish the peer review history of their article (what does this mean?). If published, this will include your full peer review and any attached files.

Reviewer #1: No

Reviewer #2: No

---

## [Author Response · Author response to Decision Letter 0]

3 Aug 2021

July 30, 2021

PLoS ONE

Dear PLoS Editor(s)

Thank you for the opportunity to revise this manuscript. I am grateful to you and the two reviewers for

your helpful comments. I think the paper is much improved and of interest to the readers of PLoS ONE!

I have submitted two versions of the manuscript. One includes the changes tracked in L A TEX. New text is in

red and removed text is marked with strike-out lines (e.g., remove this). A clean copy with these changes

also is included.

Below are items raised in the reviews and how I addressed them.

1. More emphasis was added regarding the importance of studying highly-transmissible diseases specifi-

cally on college and university campuses. In particular, the motivation for this study, further explained

in the paper, is to explore practices that increase the safety of the millions of young adults in college

and university settings. There are nearly 2700 four-year institutions in the United States, for instance,

which have approximately 10 million students (National Center for Education Statistics).

2. I removed the use of first person narrative throughout.

3. I appreciate Reviewer #1’s comment that “this paper seems a little untimely” since ”the mask mandate

for individuals who are vaccinated is no longer applicable.” In the paper’s defense, just this week the

CDC recommended vaccinated people return to wearing masks indoors. Additionally, the FDA has not

formally approved the vaccine so most institutions are unable to require students to get vaccinated.

Therefore, masks and the process of testing, isolating infections individuals and quarantining contacts

remain important and effective strategies to reduce transmission. This paper explores these in a real-

world setting. Additionally, colleges and universities are beginning another academic year with great

uncertainly about the risk of COVID-2 in academic communities, particularly with the rise of more

highly transmissible variants in the population. Finally, there certainly will be new viral pandemics in

our future and colleges and universities, with millions of students in the USA alone, need to understand

how such epidemics can spread and be controlled.

4. Reviewer #1 suggested the model could include vaccination. This is a great idea but would greatly

extend this paper and complicate the results because, undoubtedly, the additional factors (vaccination

rates, vaccination strategies, and a different vaccine efficacies) would interact with the five factors

discussed in this paper, creating an eight factor analysis. [As an aside, in the coming weeks I will

be submitting a vaccination paper and then another on accounting for the evolution of SARS-CoV-2

variants in more general settings.]

5. Reviewer #1 suggested that hand washing and social distancing might be included. Research does not

support that hand-washing significantly reduces the spread of this airborne virus and was, therefore,

not considered. Social distancing is difficult to control and the actual distances needed are not well

understood. In contrast, the data on mask efficacies and rates at which people comply in wearing

masks are well understood.

6. Reviewer #1 suggests including a diagram of the layout of the residential housing. I think this might

incorrectly imply that the model incorporates this level of detail. The model makes no assumptions

about distances between residence halls and classrooms or among student rooms and other rooms in the

same or different hallways or halls. It is assumed that the primary opportunities for viral transmission

occur inside classrooms and between students living together. The model is constrained by existing

data as represented in Figure 1.

7. Reviewer #1 asked for the number of students with the percentage mentioned (89% of the 5539

students) so I added this and referred to the percentage parenthetically.8. Reviewer #1 suggested that the “five factors” be reiterated. I did this by directing the reader to Table

2.

9. Reviewer #1 suggested that I clarify what the “higher-order interactions” are. This was added.

10. Reviewer #1 states they are confused by testing daily. This is one of the factors (testing cycles ranged

from none, once per semester, up to every day). To clarify this I added a reference to Figure 9 which

shows the different frequencies of testing and how it affects the total number of tests administered.

11. Review #2 requested clarification for the origin of parameters and so a reference to Table 2, which

contains these parameter sources, was added at the first mention of using R0 .

12. Greater detail and justification for why the SUNY Geneseo network was used has been added. Also,

additionally information on the size of the institution was included. Geneseo is a good model for this

study because it is a relatively self-contained, rural, residential, undergraduate institution. There are

several dozen graduate students that enroll in their own courses or have placements and residences off

campus (there is no on campus housing for graduate students).

13. Faculty and staff are not included in the model. It is assumed, and is the focus of this study, that

students interact primarily with each other, including in classrooms and in residence halls. This is

discussed in the new limitations paragraph.

14. Reviewer #2 asked for additional information on why this school was chosen and to provide additional

background information. This has been added to the Methods section.

15. For clarification I changed the model acronym from “SEIRIQ” to “SEIRIsol Q” to reduce confusion

that might arise from having two “I” states.

16. Reviewer #2 suggested the figure comparing college and random housing networks could be improved

by reducing the number of vertices. This is a great suggestion and was done with a smaller subset of

students. The figure caption was updated accordingly.

17. Reviewer #2 reiterates the need for clarification of purpose and limitations, which are now more

complete in the manuscript.

18. I have added a paragraph at the end of the Discussion on limitations. I think these better help the

reader interpret the results of the paper.

Thank you again for your consideration of this manuscript.

Sincerely

Gregg Hartvigsen

Biology Department

SUNY Geneseo

Geneseo, NY 14454

hartvig@geneseo.edu

2

---

## [Decision Letter · Decision Letter 1]

23 Aug 2021

Network assessment and modeling the management of an epidemic on a college campus with testing, contact tracing, and masking

PONE-D-21-17066R1

Dear Dr. Hartvigsen,

We’re pleased to inform you that your manuscript has been judged scientifically suitable for publication and will be formally accepted for publication once it meets all outstanding technical requirements.

Kind regards,

Siew Ann Cheong, Ph.D.

Academic Editor

PLOS ONE

Additional Editor Comments (optional):

Reviewers' comments:

Reviewer's Responses to Questions

**Comments to the Author**

1. If the authors have adequately addressed your comments raised in a previous round of review and you feel that this manuscript is now acceptable for publication, you may indicate that here to bypass the “Comments to the Author” section, enter your conflict of interest statement in the “Confidential to Editor” section, and submit your "Accept" recommendation.

Reviewer #1: All comments have been addressed

Reviewer #2: All comments have been addressed

2. Is the manuscript technically sound, and do the data support the conclusions?

Reviewer #1: Yes

Reviewer #2: Yes

3. Has the statistical analysis been performed appropriately and rigorously? 

Reviewer #1: Yes

Reviewer #2: Yes

4. Have the authors made all data underlying the findings in their manuscript fully available?

Reviewer #1: Yes

Reviewer #2: Yes

5. Is the manuscript presented in an intelligible fashion and written in standard English?

Reviewer #1: Yes

Reviewer #2: Yes

6. Review Comments to the Author

Reviewer #1: The revisions completed by the author have greatly improved not only the overall readability of the paper, but also have addressed the questions previously brought up. As we continue to learn more about SARS-CoV-2 and move through the COVID-19 pandemic and consistently changing mandates and guidelines from the CDC and local public health departments, information about masking, testing, quarantine and contact tracing efficiency will prove useful for colleges when deciding to implement protective health policies.

The addition and expansion of the limitation sections really strengthens the paper by address the questions that arose with the large scale roll out of vaccines, variations by demographic categories that we have seen in previous studies, and other short comings that cannot be addressed in modeling experiments.

I have minor suggestions about the grammatical structure of the paper as you will see below referenced via line number;

#2: Reframing to this demonstrate more that the information can be applicable to airborne infectious that was spurred on by the COVID-19 pandemic.

#21: "In this paper we explore..." can be changed to "This paper explores the..."

#119: Would rework this sentence to remove the term 'we'

#141: Remove "we can see that a"

#151: Remove "we can see that"

#219: Remove "In Fig 11 we can see" and put '(Fig 11)' after "subgraphs by major"

Reviewer #2: All my major comment has been addressed by the author. I have further no comments for this manuscript.

7. PLOS authors have the option to publish the peer review history of their article (what does this mean?). If published, this will include your full peer review and any attached files.

Reviewer #1: No

Reviewer #2: **Yes: **KARIKALAN NAGARAJAN

---

## [Editor Report · Acceptance letter]

31 Aug 2021

PONE-D-21-17066R1 

Network assessment and modeling the management of an epidemic on a college campus with testing, contact tracing, and masking 

Dear Dr. Hartvigsen:

I'm pleased to inform you that your manuscript has been deemed suitable for publication in PLOS ONE. Congratulations! Your manuscript is now with our production department. 

Kind regards, 

on behalf of

Dr. Siew Ann Cheong 

Academic Editor

PLOS ONE